# Abelson tyrosine-protein kinase 2 regulates myoblast proliferation and controls muscle fiber length

Jennifer K Lee, Peter T Hallock[†], Steven J Burden*

Helen L and Martin S Kimmel Center for Biology and Medicine at the Skirball Institute of Biomolecular Medicine, NYU Medical School, New York, United States

**Abstract** Muscle fiber length is nearly uniform within a muscle but widely different among different muscles. We show that Abelson tyrosine-protein kinase 2 (Abl2) has a key role in regulating myofiber length, as a loss of Abl2 leads to excessively long myofibers in the diaphragm, intercostal and levator auris muscles but not limb muscles. Increased myofiber length is caused by enhanced myoblast proliferation, expanding the pool of myoblasts and leading to increased myoblast fusion. Abl2 acts in myoblasts, but as a consequence of expansion of the diaphragm muscle, the diaphragm central tendon is reduced in size, likely contributing to reduced stamina of *Abl2* mutant mice. Ectopic muscle islands, each composed of myofibers of uniform length and orientation, form within the central tendon of *Abl2*$^{+/-}$ mice. Specialized tendon cells, resembling tendon cells at myotendinous junctions, form at the ends of these muscle islands, suggesting that myofibers induce differentiation of tendon cells, which reciprocally regulate myofiber length and orientation.

DOI: https://doi.org/10.7554/eLife.29905.001

*For correspondence:
steve.burden@med.nyu.edu

Present address: [†]Rare Muscle and Metabolic Disease, Sanofi Genzyme, Framingham, United States

Competing interests: The authors declare that no competing interests exist.

## Introduction

Skeletal muscle fibers display a wide diversity in size both within individual muscles and among different muscles. This heterogeneity in muscle fiber diameter is initiated during development and regulated throughout life, as muscle fibers adapt their size in response to metabolic demands and neural activity, prompting muscle growth or leading to muscle atrophy (*Schiaffino et al., 2013*; *Glass, 2003*). Although the mechanisms that control muscle fiber diameter are reasonably well-understood (*Schiaffino et al., 2013*; *Glass, 2003*), the mechanisms that regulate muscle fiber length and ensure a nearly common myofiber length within a muscle, but a wide diversity in length among different muscles, are largely unknown. In principle, the mechanisms for setting muscle fiber length might be regulated by the size of the myoblast pool that is available to fuse, the propensity of myoblasts to fuse, the available space that is set by the positions of fixed skeletal elements, or all of these mechanisms.

Even less is known about how signaling between muscle and tendon cells influences muscle fiber growth, differentiation and orientation. Studies in *Drosophila* suggest that tendon-like cells, prepositioned at the margins of a muscle, provide guidance cues that direct and orient myotube elongation as well as attachment sites for muscle (*Frommer et al., 1996*; *Schnorrer and Dickson, 2004*; *Wayburn and Volk, 2009*). Although there is good evidence that muscle and tendon cells exchange signals to mutually control their differentiation in *Drosophila* (*Wayburn and Volk, 2009*; *Becker et al., 1997*; *Volk and VijayRaghavan, 1994*; *Schnorrer and Dickson, 2004*), far less is known about this process in vertebrates (*Kardon, 1998*; *Schweitzer et al., 2010*). As such, it remains possible that muscle fibers provide signals to one another (*Ho et al., 1983*), influence the arrangement and properties of muscle interstitial cells (*Mathew et al., 2011*) or alter the structure

of the extracellular matrix between muscle fibers to ensure a common muscle fiber length and orientation (*Hauschka and Konigsberg, 1966*).

Here, we show that a loss of Abelson related kinase 2 (Abl2), a non-receptor tyrosine kinase, selectively in myoblasts, leads to enhanced myoblast proliferation and an increase in muscle fiber length, consistent with the idea that the size of the myoblast pool has an important influence on muscle fiber length. As a consequence of muscle expansion, the size of tendon is reduced. Moreover, we show that ectopic muscle islands, surrounded by tendon cells, form in *Abl2* heterozygous mice, yet the muscle fibers within these islands are of uniform length and orientation. These findings indicate that pre-positioned tendon cells are not essential to define the length and orientation of myofibers. Because specialized tendon cells form at the ends of these muscle islands, our results raise the possibility that a pioneering myotube induces tendon cells to organize and direct the orientation of later forming myotubes.

## Results

### Abl2 regulates muscle fiber length

The Abelson family of non-receptor tyrosine kinases, which includes Abl1 (c-Abl) and Abl2 (also known as Arg), are widely expressed and crucial mediators of growth factor and adhesion receptors that regulate cell proliferation and cytoskeletal remodeling. Although $Abl2^{-/-}$ mice survive postnatally, $Abl1^{-/-}$ mice die at birth (*Tybulewicz et al., 1991*). Thus, we studied $Abl1^{-/-}$ and $Abl2^{-/-}$ mice at embryonic day 18.5 (E18.5), one day prior to birth. We began by examining the diaphragm muscle as the muscle can be readily viewed in its entirety as a whole-mount preparation, simplifying histological analysis and providing a comprehensive view of the muscle. Muscle fiber development appeared normal in $Abl1^{-/-}$ mice, as muscle fibers extend radially from the rib cage, converge medially around the central tendon, and attach to the central tendon (*Hallock, 2011*). In the absence of Abl2, however, the diaphragm muscle fibers are excessively long, and the muscle nearly consumes the area normally occupied by the central tendon (*Figure 1A*). We measured the length of myofibers in the costal diaphragm muscle from insertion points at the ventral rib to the central tendon and found that muscle fibers in the diaphragm muscle are 1.7-fold longer in $Abl2^{-/-}$ mice than wild type littermate controls (*Figure 1B*). This expansion of muscle, is first evident at E13.5 during development of the diaphragm muscle (*Figure 1—figure supplement 1*).

To determine whether the elongated diaphragm muscle fibers grew over the normal central tendon domain or whether the central tendon was proportionally reduced in size we visualized the central tendon by using in situ hybridization to detect *scleraxis* (*Scx*) expression. We found that *Scx*, a transcription factor that is expressed in tendon precursor cells (*Schweitzer et al., 2001*), is highly expressed in tendon cells at the myotendinous junction (MTJ) (*Figure 1C*). The area of the central tendon, circumscribed by *Scx* expression, was reduced in $Abl2^{-/-}$ mice (*Figure 1D*), indicating that muscle fiber overgrowth and a reduction in size of the central tendon appear to be coordinated.

### Abl2 regulates muscle fiber length in several muscle groups in adult mice

To determine whether Abl2 had a similar role in other muscles, we analyzed intercostal, levator auris longus and hind limb muscles in adult mice. We found that muscle fibers in intercostal muscles, which ordinarily extend from one rib to the adjacent rib, were likewise excessively long and extended over multiple ribs (*Figure 2A*). Similarly, muscle fibers in the levator auris longus muscle, which normally terminate at a midline tendon boundary, appeared wavy (*Figure 2B*), suggesting that some of the increased length was accommodated by bending of the myofiber (*Figure 2B*). To determine whether the midline tendon area was reduced in the levator auris longus muscle, we analyzed transgenic mice that express GFP under the control of the *Scx* promoter in an $Abl2^{-/-}$ background (*Schweitzer et al., 2001*; *Pryce et al., 2007*). We found that the midline tendon domain, marked by GFP expression, was smaller in $Abl2^{-/-}$ than control mice (*Figure 2B*). Thus, muscle fiber overgrowth and a reduction in size of the midline tendon appear to be coordinated in the levator auris longus muscle, similar to the diaphragm muscle.

We also examined diaphragm muscles from adult mice to learn whether the muscle and tendon aberrations persisted after muscle formation. *Figure 2C* shows that the muscle remained lengthened

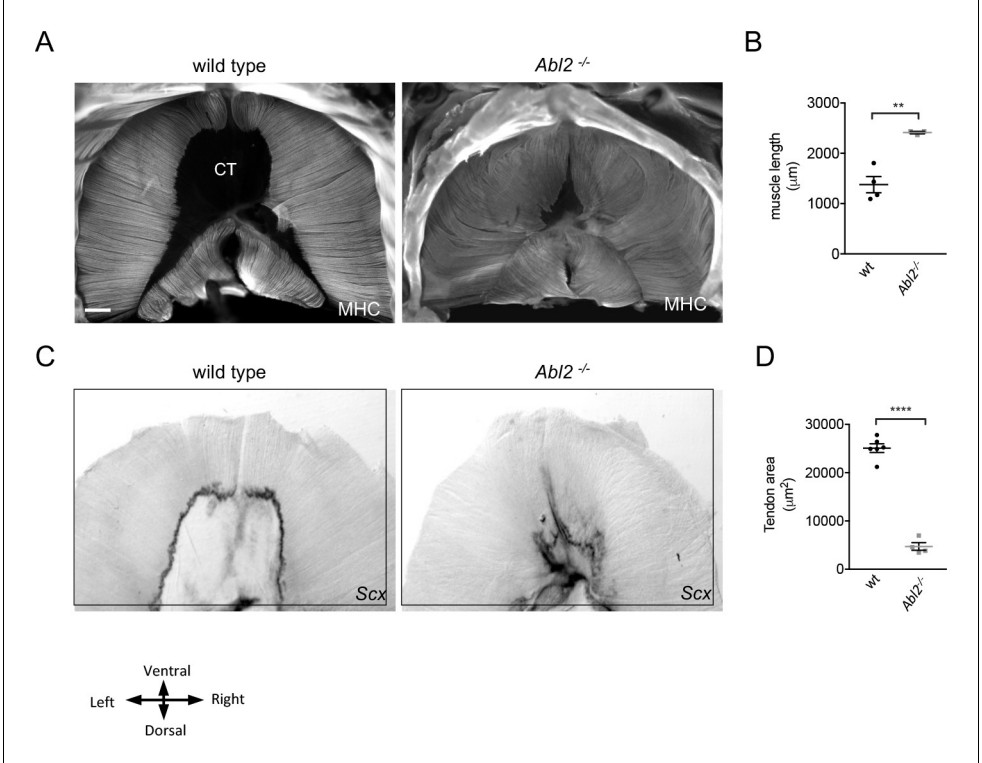

**Figure 1.** Diaphragm muscle fibers in E18.5 *Abl2* mutant mice are extraordinarily long, and the central tendon is diminished in size. Whole mounts of muscle were stained with antibodies to myosin heavy chain (MHC). (**A**) Costal muscle fibers in E18.5 embryonic diaphragm muscle normally extend from the ribcage and attach medially to the central tendon (CT). (**B**) The mean myofiber length, measured in the ventral quadrant of the costal diaphragm muscle, is ~1.7 fold longer in E18.5 *Abl2* mutant than in wild type (wt) mice. (**C,D**) The area of the central tendon, circumscribed by *Scx* RNA expression, was reduced in *Abl2*$^{-/-}$ mice. **p<0.01, ****p<0.001.

DOI: https://doi.org/10.7554/eLife.29905.002

The following figure supplement is available for figure 1:

**Figure supplement 1.** Embryonic development of *Abl2*$^{-/-}$ diaphragm.

DOI: https://doi.org/10.7554/eLife.29905.003

and the central tendon remained smaller in adult *Abl2* mutant mice (*Figure 2C*). Throughout development and in adults, myofibers are lengthened, while myofiber cross-sectional area remains normal (*Figure 2D*). In contrast, we found no evidence for an increase in muscle fiber length or cross-sectional area in hindlimb muscles (*Figure 2—figure supplement 1*). Together, these data show that Abl2 has a role in regulating muscle fiber length in many but not all muscles.

## A loss of Abl2 leads to an increase in myoblast fusion

To determine whether the elongated muscle fibers arise from an increase in myoblast fusion or an increase in cytoplasmic volume without cell fusion, we counted the number of myofiber nuclei in diaphragm muscle fibers from E18.5 wild type and *Abl2*$^{-/-}$ mice. We counted myofiber nuclei in two ways: first, we dissociated muscle fibers and counted myonuclei in individual myofibers (*Figure 3A, B*); second, we measured myonuclei number in serial cross-sections, collected from the central tendon to the ribcage of the diaphragm muscle (*Figure 3C,D*). By both methods, we found that the increase in muscle fiber length was accompanied by a proportional increase in myonuclei number.

## Abl2 acts in myoblasts to control muscle fiber growth

We performed whole mount in situ hybridization experiments to determine where *Abl2* is expressed in the diaphragm. We found that *Abl2* is highly expressed in the diaphragm muscle and absent from the central tendon in E18.5 mice (*Figure 4A,B*), suggesting that *Abl2* acts in myoblasts and/or

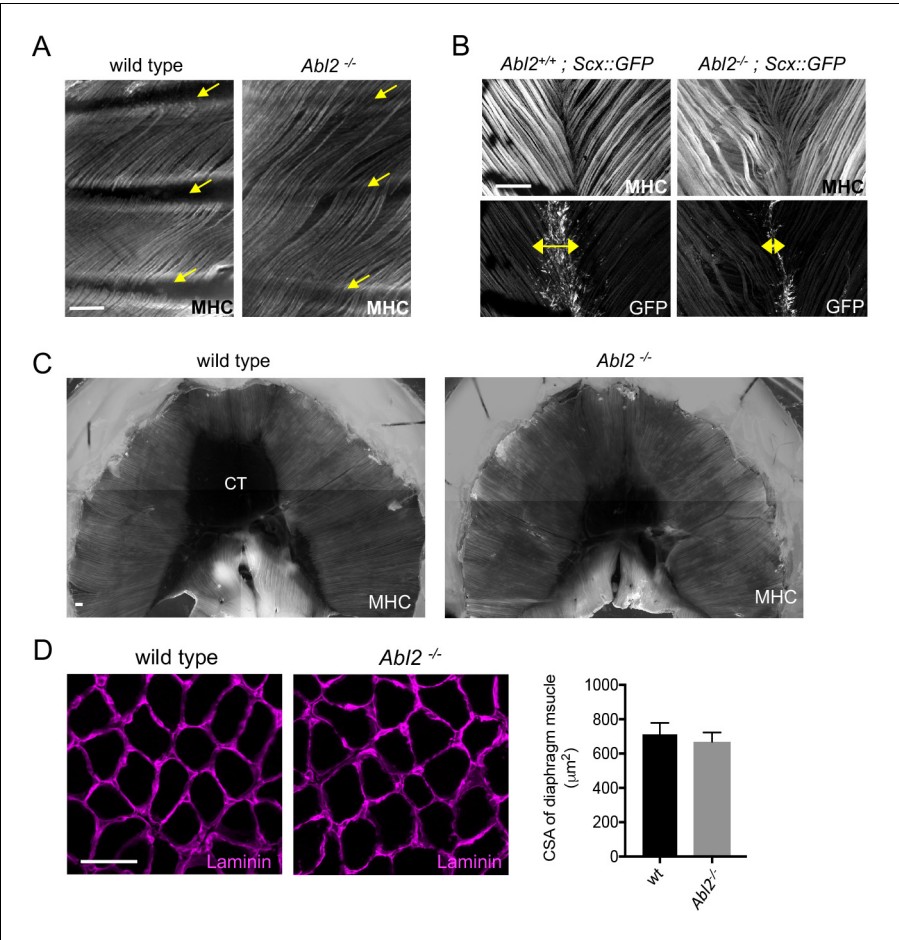

**Figure 2.** Muscle fibers are excessively long in intercostal, levator auris, and diaphragm muscles in adult *Abl2* mutant mice. Whole mounts of adult muscle were stained with antibodies to myosin heavy chain (MHC) and GFP. Cross-sections of muscle were stained with antibodies to Laminin. (**A**) Intercostal muscle fibers normally extend from one rib to the adjacent rib (arrows) but extend and cross over one or more ribs in *Abl2* mutant mice. (**B**) Muscle fibers in the levator auris muscle appear wavy in *Abl2* mutant mice. Moreover, the midline tendon, marked by *Scx::GFP*, is reduced in size (double headed arrows). (**C**) Muscle fibers in the diaphragm muscle remain longer and the central tendon is reduced in size in adult *Abl2* mutant mice. (**D**) The cross-sectional area of myofibers in adult, 8 week old, *Abl2⁻/⁻* mice is normal. Scale bar is 250 µm in A and B, and 50 µm in D.
DOI: https://doi.org/10.7554/eLife.29905.004

The following figure supplement is available for figure 2:

**Figure supplement 1.** Limb muscles appear normal in muscle length and cross-sectional area.
DOI: https://doi.org/10.7554/eLife.29905.005

myofibers to regulate muscle growth. To investigate whether Abl2 is expressed in myoblasts or multinucleated myotubes, we analyzed Abl2 protein expression during muscle differentiation in the C2C12 muscle cell line. We found that Abl2 expression is high in myoblasts and declines during differentiation (*Figure 4—figure supplement 1*). These results indicate that myoblasts, rather than myotubes, are the major source of Abl2 expression in muscle.

To examine the site of action of *Abl2* during myogenesis, we obtained mice bearing an *Abl2* allele with *loxP* sites and generated conditional *Abl2* mutants using mice that express Cre recombinase in pre-migratory myogenic precursor cells (*Pax3^Cre*), committed myoblasts (*Myod^iCre*) or multinucleated myofibers (*Mck::Cre*) (*Engleka et al., 2005*; *Relaix et al., 2006*; *Kanisicak et al., 2009*; *Braun et al., 1994*; *Brüning et al., 1998*).

Selective inactivation of *Abl2* in pre-migratory myogenic precursor cells (*Abl2^f/−; Pax3^cre*) led to muscle defects like those found in *Abl2⁻/⁻* mice (*Figure 4C,D*). Likewise, inactivation of *Abl2* in

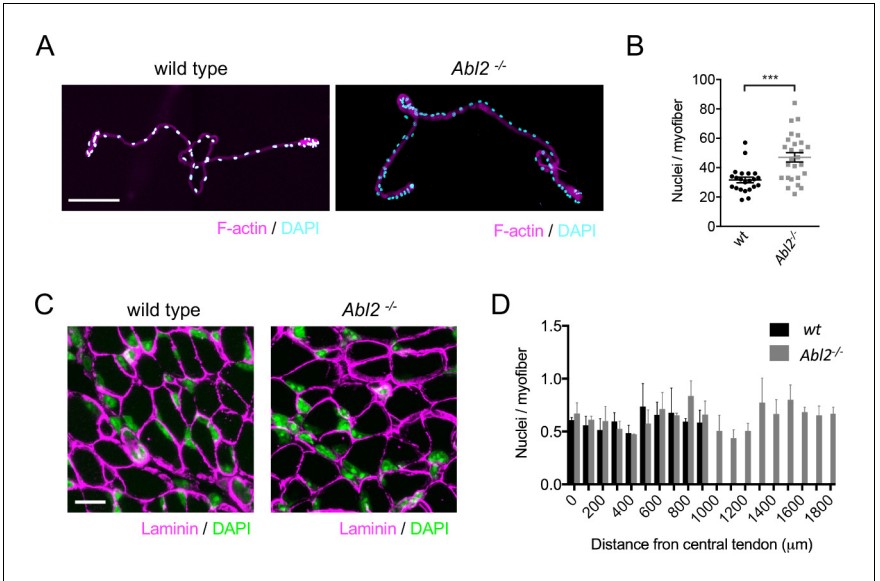

**Figure 3.** The increase in muscle fiber length is accompanied by an increase in myonuclei number. (**A**) Single fibers dissociated from the ventral quadrant of the costal diaphragm muscle were stained with Phalloidin to label F-actin and DAPI to label nuclei. (**B**) The scatter plot shows the number of nuclei per myofiber for individual dissociated myofibers, dissected from three individual *Abl2* mutant and three wild type (wt) mice, as well as the mean ±SEM. (**C**) Representative images from serial cross sections of the diaphragm muscle, stained for DAPI to label nuclei and with antibodies to Laminin to outline muscle fibers. (**D**) The number of myonuclei in each section was divided by the number of muscle fibers. Myonuclei are similarly spaced along the entire length of muscle from wild type and *Abl2* mutant mice. The graph shows the mean ±SEM per field of view in serial cross sections taken every 100 μm from the central tendon to the ribcage of 3 wild type and 3 *Abl2* mutant mice. Scale bar is 150 μm in A and 10 μm in C. ***$p < 0.005$.

DOI: https://doi.org/10.7554/eLife.29905.006

committed myoblasts (*Abl2^{f/−}; Myod^{icre}*) caused aberrations in muscle development identical to those found in *Abl2^{−/−}* mice (**Figure 4C,E**). In contrast, inactivation of *Abl2* in myofibers (*Abl2^{f/−}; Mck::Cre*) did not cause a muscle phenotype (**Figure 4F**). These findings indicate that Abl2 is required in myoblasts rather than multinucleated myotubes to regulate muscle and tendon growth.

## *Abl2* mutant mice display impaired motor endurance

Diaphragm muscle fibers converge and attach to the central tendon. During respiration, the cyclical cycles of muscle contraction cause the central tendon to shift from a relaxed to a taut sheet, causing changes in the volume and pressure within the thoracic cavity and promoting exchange of gases critical for respiration. To determine whether replacement of the central tendon with muscle might alter the elasticity and stiffen the diaphragm, leading to respiratory deficits, we assessed respiratory function by whole body plethysmography, which measures basal respiration in resting mice. We did not detect a difference in resting breathing rates (data not shown).

We reasoned that respiratory deficits may be unapparent during rest and only revealed during exercise. To assess respiratory function during exercise, mice were examined on a running wheel task, and the distance traveled over a 20 hr period was measured. We found that *Abl2* mutant mice ran less than wild type littermates (**Figure 5A**). Because limb muscle development and limb strength were normal in *Abl2* mutant mice (**Figure 2—figure supplement 1**, **Figure 5B**), the decrease in running distance suggests that replacement of the central tendon with muscle in the diaphragm impaired respiration when challenged in an endurance task.

*Abl2* mutant mice display several behavioral abnormalities, including deficits in coordination, startle response and aggression, which are attributed to a role for Abl2 in the central nervous system (**Koleske et al., 1998**). We therefore considered that a loss of Abl2 from the nervous system may be responsible for the reduced running performance of *Abl2* mutant mice. To determine whether the

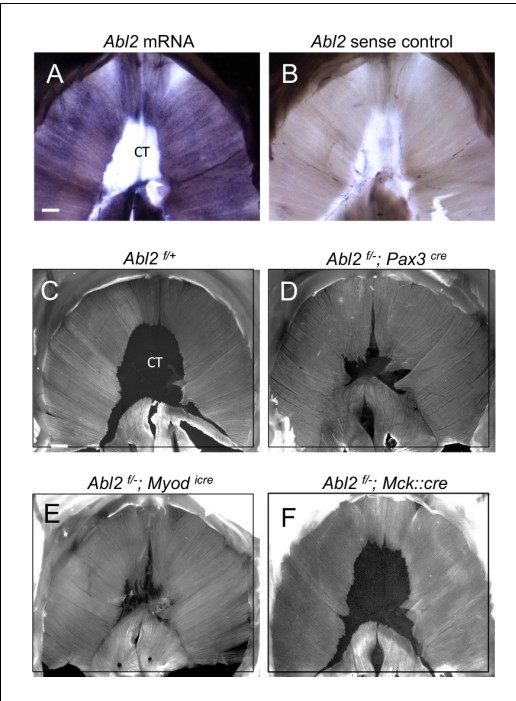

**Figure 4.** Abl2 acts in myoblasts to regulate muscle development. (A,B) *Abl2 mRNA* is highly expressed in the muscle but not within the central tendon (CT) of the diaphragm. (C) Muscle and tendon development appear normal in *Abl2*$^{flox/+}$ control mice (*Abl2*$^{f/+}$). Muscle length is increased and tendon size is reduced by conditionally inactivating *Abl2* in (D) muscle precursors (*Abl2*$^{f/-}$; *Pax3*$^{cre}$), or (E) committed myoblasts (*Abl2*$^{f/-}$; *Myod*$^{icre}$). (F) Muscle and tendon development appear normal by inactivating *Abl2* in mature myotubes (*Abl2*$^{f/-}$; *Mck::cre*). Scale bars are 500 μm in A and C.

DOI: https://doi.org/10.7554/eLife.29905.007

The following figure supplement is available for figure 4:

**Figure supplement 1.** Abl2 expression is enriched in proliferating myoblasts.

DOI: https://doi.org/10.7554/eLife.29905.008

reduced stamina of *Abl2* mutant mice was due to a loss of muscle-derived Abl2 we studied the running performance of mice lacking *Abl2* selectively in muscle (*Abl2*$^{f/-}$; *Myod*$^{icre}$ mice). Like *Abl2* mutant mice, the *Abl2* muscle-conditional mutant mice displayed reduced stamina on the running wheel (*Figure 5A*). Together, these results suggest that replacement of tendon with muscle in the diaphragm impairs respiration leading to impaired physical endurance.

We considered the possibility that the decreased endurance was due to a failure to develop slow, fatigue-resistant muscle fibers (*Agbulut et al., 2003*). We therefore analyzed additional features of muscle differentiation in *Abl2*$^{f/-}$; *Myod*$^{icre}$ adult diaphragm muscle and found that the number of slow myofibers was normal (*Figure 5C*). Therefore, the decreased stamina of muscle conditional *Abl2* mutant mice cannot be attributed to a failure to develop slow muscle fibers. Further, muscle fibers in *Abl2*$^{-/-}$ adult mice displayed ultrastructural features characteristic of skeletal muscle, as the organization of sarcomeres and muscle-tendon attachment sites appeared normal (*Figure 5—figure supplement 1*). Thus, the decreased physical performance of *Abl2* mutant mice is not caused by a failure to build the contractile apparatus and is more likely caused by the replacement of the elastic central tendon with less-tensile skeletal muscle.

## Abl2 negatively regulates myoblast proliferation

Analysis of single muscle fibers demonstrated that increased muscle fiber length was accompanied by an increase in myoblast fusion (*Figure 4A,B*). Examination of *Abl2* conditional mutants revealed that a loss of Abl2 from myoblasts was responsible for the lengthened myofibers (*Figure 5*). From these data, we hypothesized that the increased myoblast fusion in *Abl2*$^{-/-}$ mice might arise from an expanded pool of myoblasts produced by increased myoblast proliferation. To test this idea, we quantified the proliferation rate of myoblasts by measuring EdU (5-ethynyl-2′-deoxyuridine) incorporation in MyoD$^+$ myoblasts in wild type and *Abl2* mutant embryos. At E14, the percentage of myoblasts that incorporated EdU was greater in *Abl2* mutant mice than wild type mice (*Figure 6A,B*), indicating that a loss of Abl2 leads to increased proliferation of myoblasts in vivo.

Further, we measured the proliferation rate of myoblasts in primary cultures of diaphragm muscles from E18.5 *Abl2* mutant and wild type mice. A similar number of MyoD$^+$ myoblasts were isolated from wild type and *Abl2* mutant mice at E18.5 (*Figure 6C,D*). After growing cells at low density for 36 hr, the cultures were pulse-labeled with EdU for 1 hr, fixed, and stained for MyoD to identify proliferating myoblasts. We found that a higher percentage of A*bl2* mutant myoblasts incorporated EdU than wild type control myoblasts ($69.7 \pm 4.1\%$ and $44.1 \pm 2.6\%$ mean±SEM respectively) (*Figure 6E*). In contrast, cells that did not express MyoD, such as fibroblasts or tendon cells, did not differ in EdU incorporation (*Figure 6F*). Thus, Abl2 negatively regulates proliferation and expands

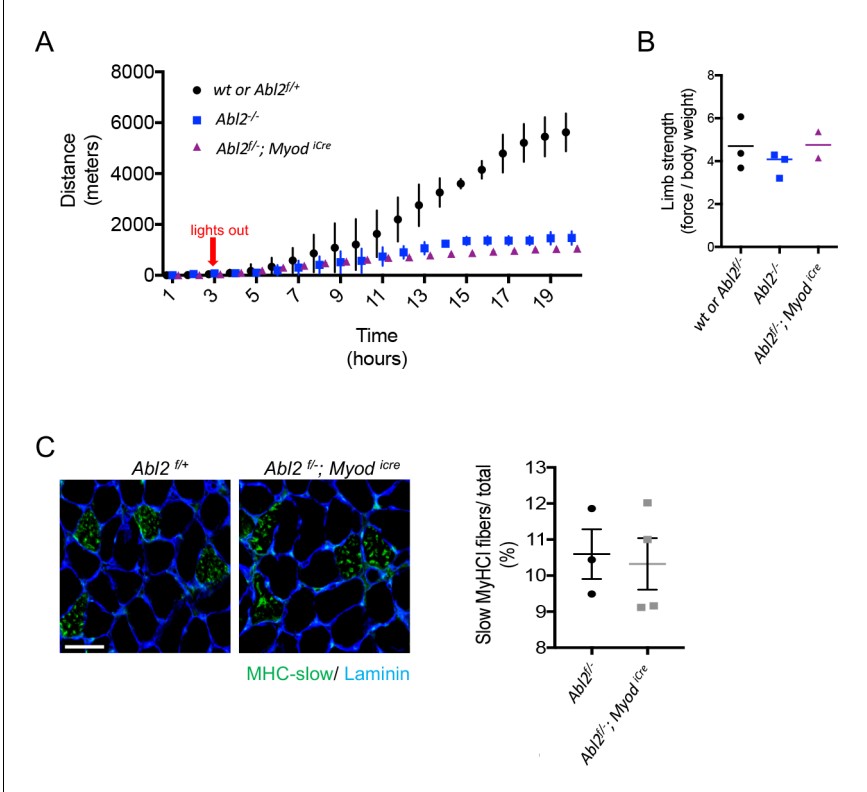

**Figure 5.** The exercise endurance of *Abl2* null and *Abl2* muscle-conditional mutant mice is impaired. (**A**) During a twenty-hour period, *Abl2*$^{-/-}$ and *Abl2*$^{f/-}$; *Myod*$^{iCre}$ conditionally mutant mice run less than wild type (wt) mice. (**B**) The limb strength of *Abl2*$^{-/-}$, *Abl2*$^{f/-}$; *Myod*$^{iCre}$ and wild type mice are similar. The graph and scatter plot show the values for individual mice and the mean values together with the SEMs. All *Abl2* null and conditional *Abl2* mutant mice were tested at 18 weeks with littermate controls. (**C**) Cross-sections of the diaphragm muscle were stained with Laminin and myosin heavy chain Type I (MHC-slow) to identify slow-twitch muscle fibers. The number of slow-twitch fibers is normal in *Abl2* mutant mice. The scatter plot shows the percentage of myofibers in cross-sections of muscle that were stained by antibodies to MHC-I. The mean values and SEMs from three control (*Abl2*$^{f/+}$) and four muscle-conditional *Abl2* mutant mice (*Abl2*$^{f/-}$; *Myod*$^{iCre}$) are shown. Scale bar is 50 µm.

DOI: https://doi.org/10.7554/eLife.29905.009

The following figure supplement is available for figure 5:

**Figure supplement 1.** Ultrastructural appearance of muscle-tendon junctions and muscle fibers is normal in *Abl2* mutant mice.

DOI: https://doi.org/10.7554/eLife.29905.010

---

the pool of myoblasts but not of other cell types in muscle tissue. Further, we compared EdU incorporation among wild type, *Abl2*$^{+/-}$ and *Abl2* mutant cells and found that *Abl2*$^{+/-}$ myoblasts proliferated at a rate that was intermediate between wild type and *Abl2* homozygous mutant myoblasts (*Figure 6G*). We also examined proliferation of myoblasts by double-staining with antibodies to MyoD and phospho-histone H3, a marker specific for cells undergoing mitosis, which revealed an increased number of mitotic myoblasts in *Abl2*$^{-/-}$ cultures compared to wild type littermate cultures (*Figure 6H,I*). These data indicate that the longer myofibers in *Abl2*$^{-/-}$ mice arise from enhanced proliferation of *Abl2*$^{-/-}$ myoblasts.

## Enhanced myoblast proliferation in *Abl2*$^{-/-}$ mice leads to enhanced myoblast fusion

To determine whether increased muscle fusion was caused by enhanced proliferation of *Abl2* mutant myoblasts, we compared muscle fusion under conditions that were either permissive or non-permissive for myoblast proliferation. Primary muscle cells from *Abl2*$^{-/-}$ mice were propagated under proliferative conditions for 48 hr and then switched to differentiation medium for 48 hr. Under these

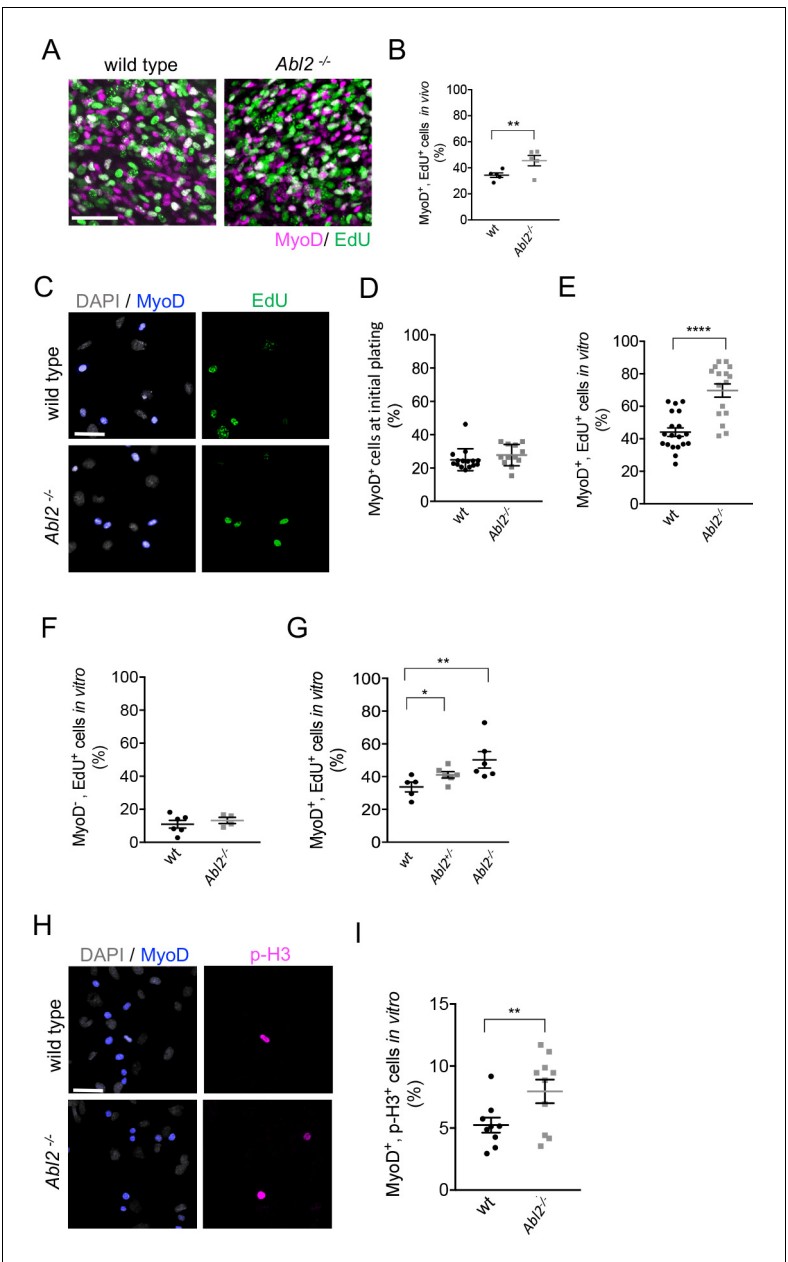

**Figure 6.** Myoblast proliferation is enhanced by a loss of *Abl2*. (A,B) In vivo labeling of E13.5-E14.5 diaphragm muscles shows that EdU incorporation is greater in *Abl2*$^{-/-}$ myoblasts than wild type (wt) myoblasts. (C) Representative images of cultured diaphragm cells stained with MyoD to label myoblasts, DAPI to label all cells and EdU to label proliferating cells. (D) At initial plating, a similar number of MyoD$^+$ cells are isolated from *Abl2*$^{-/-}$ and wild type mice. (E) Cultured MyoD$^+$ myoblasts from *Abl2*$^{-/-}$ diaphragm muscles showed increased EdU incorporation. (F) In contrast, non-muscle cells (MyoD$^-$) from *Abl2* mutant and wild type mice proliferate at similar rates. (G) *Abl2* heterozygous myoblasts proliferated at a rate that was intermediate between wild type and *Abl2* homozygous mutant myoblasts. (H) Representative images of cultured diaphragm cells stained with MyoD to label myoblasts, DAPI to label all cells, and phospho-Histone H3 (pHH3), a marker for mitotic cells. (I) MyoD$^+$ myoblasts, cultured from the diaphragm muscle of *Abl2*$^{-/-}$ mice, showed a greater percentage of mitotic figures than myoblasts isolated from wild type mice. *p=0.1, **p<0.05, ****p<0.001.
DOI: https://doi.org/10.7554/eLife.29905.011

The following figure supplement is available for figure 6:

**Figure supplement 1.** The number of Pax7$^+$ cells is increased in proportion to the increased length of myofibers.
DOI: https://doi.org/10.7554/eLife.29905.012

conditions, myotube formation was greater in cultures from $Abl2^{-/-}$ than wild type mice, leading to a greater fusion index calculated by the ratio of fused myoblasts to total myoblasts (*Figure 7A,B*). Consistent with these data, *Abl2* mutant myotubes contained more nuclei than wild type myotubes (*Figure 7C*).

Although the increased proliferation of *Abl2* mutant myoblasts could be solely responsible for the enhanced myoblast fusion, it remained possible that Abl2 also has a direct role in the mechanics of myoblast fusion. We therefore sought to determine whether the increase in myotube size was due solely to an increase in myoblast proliferation, thereby increasing the number of myoblasts available for fusion, or whether *Abl2* mutant myoblasts also had an enhanced propensity for cell fusion. As such, we plated primary cells at near confluent density into non-proliferative, differentiation conditions and scored for myoblast fusion two days later. Under these conditions, which are non-permissive for proliferation, wild type and $Abl2^{-/-}$ cultures had similar fusion indices (*Figure 7D,E*). Together, these data indicate that the longer myofibers in $Abl2^{-/-}$ mice arise from enhanced proliferation of $Abl2^{-/-}$ myoblasts, increasing the size of the available myoblast pool, rather than an enhanced propensity of myoblasts to fuse.

Because a similar number of MyoD[+] myoblasts are isolated from wild type and *Abl2* mutant mice at E18.5 (*Figure 6D*), our data suggest that the increase in myoblast proliferation is accommodated and balanced by rapid myoblast fusion, so that the myotubes in *Abl2* mutant mice accrue more nuclei and elongate.

Pax7-positive satellite cells are set-aside as a quiescent population of muscle stem cells during fetal development and are responsible for postnatal myofiber growth, homeostasis, and regeneration (*Relaix et al., 2005*; *Kassar-Duchossoy et al., 2005*; *Keefe et al., 2015*; *Lepper and Fan, 2010*). We wondered whether the expansion and fusion of myoblasts in *Abl2* mutant mice might occur at the expense of establishing a Pax7-positive satellite cell population. We counted Pax7-

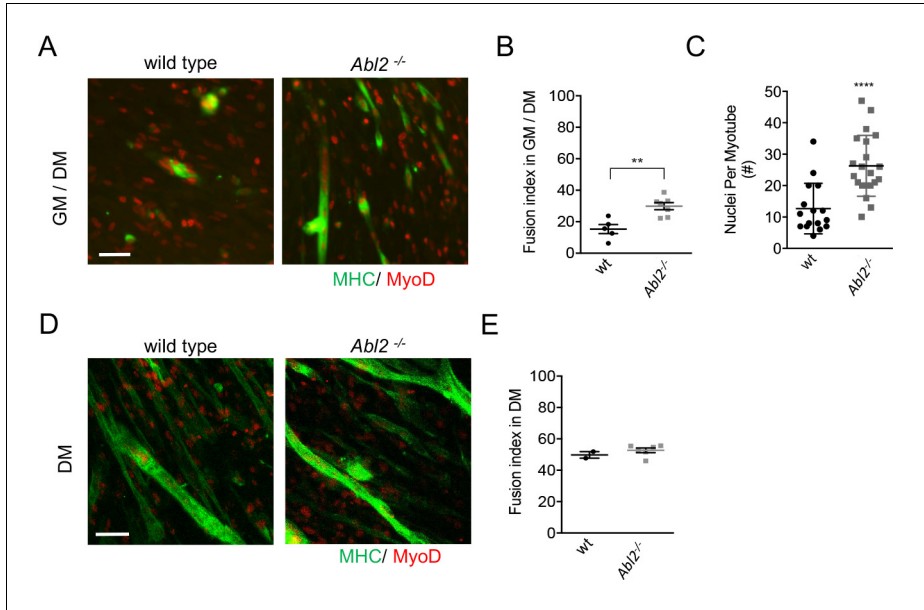

**Figure 7.** Enhanced myoblast proliferation in $Abl2^{-/-}$ mice leads to enhanced myoblast fusion. Primary cultures from the diaphragm muscle of E18.5 mice were stained with antibodies to myosin heavy chain (MHC) to label differentiated muscle fibers and MyoD to label myoblasts. (**A**) Representative images of diaphragm muscle cells, which were proliferated for 2 days before a switch to differentiation medium. (**B**) Under these conditions, *Abl2* mutant myoblasts displayed enhanced myotube formation. (**C**) The number of nuclei per myotube was quantified. Differentiated *Abl2* mutant myotubes incorporated more nuclei per myotube. (**D**) Representative images of *Abl2* mutant myoblasts, which were plated at confluent density and directly into differentiation media. (**E**) Confluent cultures, which did not have an opportunity to proliferate, formed myotubes like wild type myoblasts. Scale bars are 50 µm in A,I, and L. \*\*p<0.01, \*\*\*\*p<0.001.
DOI: https://doi.org/10.7554/eLife.29905.013

positive muscle satellite cells in serial cross-sections of *Abl2* mutant and control mice and found that Pax7-positive satellite cells were evenly distributed along the length of the diaphragm muscle fibers, and that total satellite cell number increased in *Abl2*$^{-/-}$ mice in proportion to the increase in muscle length (*Figure 6—figure supplement 1*). These findings indicate that increased muscle length did not occur at the expense of establishing the satellite cell pool and that Pax7-positive satellite cell numbers are established in proportion to muscle size.

## Ectopic muscle islands form within the central tendon of *Abl2*$^{+/−}$ mice

Mice that are heterozygous for *Abl2* also display a muscle phenotype. In wild type mice, the central tendon of the diaphragm is devoid of muscle, whereas we observed ectopic muscle islands in the central tendon of *Abl2*$^{+/−}$ mice (*Figure 8A*). The number of ectopic muscle islands as well as the number of muscle fibers within each island were variable (*Figure 8B*), but individual muscle fibers within an island were uniform in length and aligned with respect to one another. These findings suggest that the muscle fibers within an island coordinate their orientation and length, in a manner that is independent of contact with a preexisting specialized tendon cell border (*Volk and VijayRaghavan, 1994*).

Although the location of the ectopic islands within the central tendon was variable, the orientation of the ectopic islands was biased to align with the nearby diaphragm muscle (*Figure 8—figure supplement 1*). This alignment of the ectopic muscle islands with the nearby diaphragm muscle raises the possibility that myoblasts, which are aligned with developing myotubes in the diaphragm

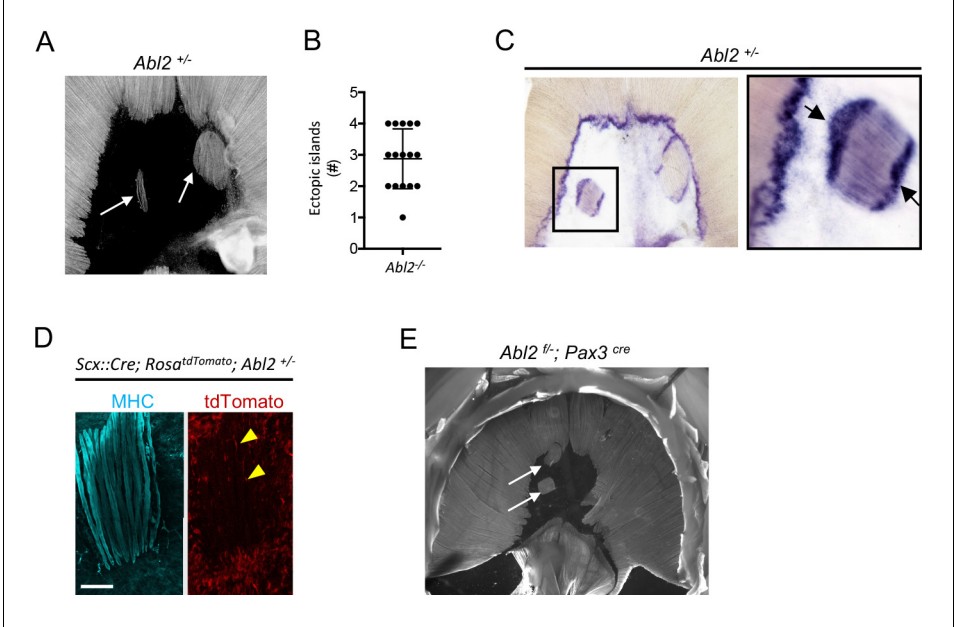

**Figure 8.** Ectopic muscle islands, which are found in the central tendon of mice that are heterozygous for *Abl2*, induce tendon cell differentiation. (A,B) Whole mount diaphragms, stained with antibodies to myosin heavy chain (MHC), revealed 1 to 4 ectopic islands in the central tendon of *Abl2*$^{+/−}$ mice. (C) *Scx* RNA expression is enhanced at the ends of muscle fibers in the ectopic islands (black arrows) as well as at the normal MTJ in the diaphragm muscle. (D) Lineage-tracing experiments in *Abl2*$^{+/−}$ mice reveal that tendon cells, marked by tdTomato (arrowheads), intercalate between MHC-stained muscle fibers within the ectopic islands but do not contribute to myofibers. (E) Whole mount images reveal ectopic muscles in the central tendon of muscle conditional *Abl2* heterozygous mice (*Abl2*$^{f/+}$; *Pax3*$^{cre}$). Scale bar is 250 μm in D.
DOI: https://doi.org/10.7554/eLife.29905.014

The following figure supplement is available for figure 8:

**Figure supplement 1.** The orientation of ectopic muscle fibers correlates with the orientation of myofibers in the nearby main costal diaphragm muscle.
DOI: https://doi.org/10.7554/eLife.29905.015

muscle, retain this orientation as they aberrantly migrate, proliferate and differentiate in the central tendon.

## Tendon cells become specialized when juxtaposed with ectopic muscle islands in $Abl2^{-/-}$ mice

We found that tendon cells at MTJs express high levels of *Scx* (*Figure 1C*). We examined whether tendon cells, positioned at the novel contact sites with the ectopic muscle islands, became similarly specialized. We found that *Scx* RNA expression was enhanced in tendon cells that are juxtaposed to the ends of the ectopic muscle islands (*Figure 8C*). These findings suggest that muscle fibers, even when displaced, provide signals to tendon cells, contributing to specialized features of tendon cells at MTJs.

## Tendon cell precursors do not contribute to muscle in ectopic islands of $Abl2^{+/-}$ mice

We wondered whether the muscle cells in the ectopic muscle islands formed from bona-fide muscle precursors or whether tendon cell precursors had altered their cell fate to form the ectopic muscle islands in $Abl2^{+/-}$ mice. We used *Scx::Cre* mice and $Rosa^{dTomato}$ reporter mice, which harbor a loxP-flanked STOP cassette preventing tdTomato transcription except in cells that express Cre recombinase, to trace the lineage of *Scx*-expressing tendon precursor cells. We found that *Scx*-expressing cells contributed to tendon cells surrounding and intercalated between muscle fibers but did not contribute to the muscle fibers within the ectopic muscle islands (*Figure 8D*). These findings are inconsistent with the idea tendon cell precursors switched their fate and converted to muscle to form the ectopic muscle islands in $Abl2^{+/-}$ mice. To determine whether a reduction in muscle expression of Abl2 was responsible for formation of the ectopic muscle islands, we inactivated one copy of *Abl2* selectively in muscle precursors. *Figure 8E* shows that ectopic muscle islands form in $Abl2^{f/+}$; $Pax3^{cre}$ mice, indicating that a loss of Abl2 in myogenic cells is responsible for formation of the ectopic muscle islands.

## Discussion

We have uncovered a novel role for Abl2 in regulating muscle fiber length in mammals. We provide evidence that a loss of Abl2 function in myoblasts leads to enhanced myoblast proliferation and an expansion of the myoblast pool during embryonic development, leading to increased myoblast fusion and elongated muscle fibers.

It is striking that a loss of Abl2 leads to an increase in muscle length without a hypertrophic increase in muscle fiber diameter. Our in vivo and in vitro data describe an enhancement of myoblast proliferation and fusion during primary/embryonic myogenesis at E13.5 and during secondary/fetal myogenesis at E18.5 (*Biressi et al., 2007*), which are developmental stages when myoblasts fuse preferentially to the ends of developing muscle fibers (*Zhang and McLennan, 1995*). This mode of iterative nuclear accrual at ends of developing muscle fibers may favor longitudinal growth of muscle during embryonic development. It is possible that increased myoblast proliferation during early embryonic development favors longitudinal lengthening of muscle fibers rather than an increase in myofiber diameter. In contrast, in adult myofibers, increased satellite cell proliferation may lead to satellite cell fusion all along the myofiber, favoring an increase in muscle fiber diameter and muscle hypertrophy.

Abl kinases are up-regulated in chronic myeloid leukemia, promoting cell proliferation. Abl2, however, attenuates cell proliferation in solid tumors, suggesting that Abl kinases can negatively or positively regulate proliferation, depending on cellular context (*Greuber et al., 2013*; *Gil-Henn et al., 2013*). Here, we provide evidence that Abl2 attenuates myoblast proliferation, limiting expansion of the myoblast pool and muscle growth. We do not know what lies upstream of Abl2 in myoblasts to control Abl2 activity, but it seems likely that multiple signals and steps modulate Abl2 activity to regulate myoblast proliferation and muscle growth.

Muscle precursors migrate to their final destinations using two distinct mechanisms. Body wall muscles, including the intercostal, vertebral, and abdominal muscles, extend and expand to envelop the body as a continuous outgrowth of the myotome (*Christ et al., 1983*; *Brand-Saberi and Christ, 1999*). In contrast, development of limb and diaphragm muscles requires delamination of muscle

precursor cells from the dermomyotome, which engage in long-range migration to their target sites, where they proliferate, differentiate and fuse (*Noden, 1983*; *Dietrich, 1999*). Although intercostal and diaphragm muscles develop through non-migratory and migratory mechanisms, respectively, Abl2 has a role in governing muscle length in both muscles, indicating that Abl2 does not act primarily, if at all to regulate myoblast migration.

Abl2, however, does not regulate muscle fiber length in limb muscles. There is precedence for differing requirements for the development of diaphragm and limb muscles: although both limb and diaphragm muscles are derived from Lbx1-expressing precursors, migration of muscle precursors to the limb, but not the diaphragm, is perturbed in *Lbx1* mutant mice (*Gross et al., 2000*). Thus, the development of diaphragm and limb muscles differ in their requirement for Abl2, as well as Lbx1.

Further, the role of Abl2 in some but not all muscles may be associated with the location, cellular environment or function of the muscle. Elongated muscle fibers are found in intercostal muscles, which attach to tendons that anchor the muscle to bone. Elongated muscle fibers are also found in muscles that are joined by tendons that interconnect two muscle segments, such as the central tendon of the diaphragm and the midline tendon between right and left levator auris longus muscles. However, muscles that attach to bone by force-transmitting tendons, such as those in the limb, remain unaffected by mutations in *Abl2* (*Murchison et al., 2007*).

In *Drosophila*, there is good evidence that muscle cells provide signals to tendon cells to induce their differentiation. Muscle cells produce Vein, a Neuregulin-like protein, which binds to the EGFR in adjacent tendon cells to mutually promote tendon cell differentiation (*Yarnitzky et al., 1997*; *Volk, 1999*). Activation of the EGFR in tendon cells enhances expression of Stripe and Slit, which signals back to Robo in muscle cells, contributing to muscle-specific tendon attachment (*Kramer et al., 2001*; *Volohonsky et al., 2007*). Although there is evidence supporting the idea that muscle and tendon cells signal to one another in vertebrates, these studies have largely focused upon TGF- and FGF-dependent signaling between the developing myotome and the adjacent sclerotome, where muscle is required for tendon formation (*Brent et al., 2003*; *Schweitzer et al., 2010*). Apart from this early axial signaling system, signaling between muscle and tendon in mammals has been not been fully explored. Unlike the formation of axial tendons, the induction of head and limb tendons does not depend upon muscle, although muscle and muscle contraction play a role in full tendon differentiation (*Gaut and Duprez, 2016*).

We provide evidence that vertebrate muscle cells provide signals that are instructive for tendon cell differentiation, as myotubes that form as islands surrounded by tendon cells, induce tendon cell differentiation at the ends of the muscle fibers. Moreover, the myotubes within these islands are uniform in length and orientation, an unexpected finding if pre-positioned and pre-specialized tendon cells were required to organize muscle length and orientation. Instead, our findings raise the possibility that muscle pioneers, akin to those described in invertebrates (*Ho et al., 1983*) may induce features of tendon cell differentiation, which subsequently lead to reciprocal signaling from tendon cells to muscle and contribute to an arrest of myotube growth, limiting myotube length and organizing the parallel alignment of myotubes.

Muscle islands are apparent in muscle conditional $Abl2^{+/-}$ mice, indicating that a decrease in Abl2 expression in myoblasts is responsible for formation of the ectopic islands. We do not know whether enhanced myoblast proliferation plays a role in formation of the ectopic muscle islands in $Abl2^{+/-}$ mice. The proliferation rate of $Abl2^{+/-}$ myoblasts is intermediate between wild type and $Abl2^{-/-}$ myoblasts, so enhanced proliferation of $Abl2^{+/-}$ myoblasts may contribute to the formation of ectopic muscle islands. We also considered the possibility that Abl2 might limit myoblast migratory behavior, preventing promiscuous migration of $Abl2^{+/-}$ myoblasts into the central tendon, but live imaging studies in cell culture failed to detect aberrant migratory behavior of *Abl2* mutant myoblasts (data not shown). In addition, myoblasts that inadvertently migrate into the central tendon domain may normally be eliminated, and this elimination process may be attenuated if Abl2 expression in myoblasts is reduced.

The ectopic islands were oriented to align with the nearby diaphragm muscle. The oriented alignment of the ectopic muscle islands with the nearby diaphragm muscle raises the possibility that *Abl2* mutant myoblasts, which migrated along the developing myotubes of the diaphragm muscle in a directed manner, aberrantly continued their oriented migration into the central tendon to form the ectopic muscle islands. Alternatively, contraction of the diaphragm muscle, causing oriented deformation of the central tendon, may assist in aligning myotubes in the ectopic islands.

Regardless of the mechanisms responsible for forming the ectopic islands, the organization of the myotubes within each island reveal mechanisms not previously understood and appreciated for aligning and setting the length of myofibers. Notably, although the muscle islands are enveloped by tendon cells, the myotubes are uniform in length and orientation, demonstrating that pre-positioned, specialized tendon cells are not essential to organize these features of muscle development. Myotubes may signal to one another to establish a common orientation and length. However, our results raise the possibility that specialized tendon cells, once induced by a developing, pioneer myotube, signal back to later arriving myotubes and contribute to their orderly arrangement.

# Materials and methods

## Key resources table

| Reagent type (species) or resource | Designation | Source or reference | Identifiers | Additional information |
|---|---|---|---|---|
| gene (*Mus musculus*) | Abl2 | The Jackson Laboratory | RRID:MGI:2653897 | |
| gene (*Mus musculus*) | Myod iCre | The Jackson Laboratory | RRID:IMSR_JAX:014140 | |
| gene (*Mus musculus*) | Mck-Cre | The Jackson Laboratory | RRID:IMSR_JAX:006475 | |
| gene (*Mus musculus*) | Pax3 Cre | The Jackson Laboratory | RRID:IMSR_JAX:005549 | |
| gene (*Mus musculus*) | Rosa 26 LacZ | The Jackson Laboratory | RRID:IMSR_JAX:003474 | |
| gene (*Mus musculus*) | Scx::Cre | MGI:5317938 | n/a | Ronen Schweitzer lab |
| gene (*Mus musculus*) | Scx::GFP | PMID: 17497702 | n/a | Ronen Schweitzer lab |
| strain, strain background (*Mus musculus*) | C57BL/6J | The Jackson Laboratory | RRID:IMSR_JAX000664 | |
| genetic reagent (*Mus musculus*) | Targeting Vector: Abl2tm1a(EUCOMM)Hmgu | EUCOMM | RRID:SCR_003104 | European Conditional Mouse Mutagenesis Program |
| cell line (*Mus musculus*) | C2C12 skeletal muscle cells | Burden lab/ATCC | Cat# CRL-1772, RRID:CVCL_0188 | |
| antibody | Rabbit anti-Abl2 | Proteintech Group | Cat# 17693–1-AP RRID:AB_2289025 | 1:1000 |
| antibody | Goat anti-Arg (C-20) | Santa Cruz Biotechnology | Cat# sc-6356 RRID:AB_2221106 | 1:200 |
| antibody | Rabbit anti-β-Actin (13E5) | Cell Signaling Technology | Cat # 4970 | 1:1000 |
| antibody | Mouse anti-Myosin (MY-32) | Sigma-Aldrich | Cat# M7523 RRID:AB_260649 | 1:1000 |
| antibody | Chicken anti-GFP | Abcam | Cat# ab92456 RRID:AB_10561923 | 1:3000 |
| antibody | Mouse anti-Myosin (NOQ7.5.4D) | Sigma-Aldrich | Cat# M8421 RRID:AB_477248 | 1:1000 |
| antibody | Rabbit anti-Laminin | Sigma-Aldrich | Cat# L9393 RRID:AB_477163 | 1:1000 |
| antibody | Mouse anti-Pax7 | Santa Cruz Biotechnology | Cat# sc-81648 RRID:AB_2159836 | 1:500 (bioreactor) |
| antibody | Rabbit anti-MyoD C-20 | Santa Cruz Biotechnology | Cat# sc-304 RRID:AB_631992 | 1:1000 |
| antibody | Mouse anti-MyoD (5.8A) | ThermoFisher Scientific | Cat# MA1-21816 RRID:AB_560242 | 1:1000 |
| antibody | Donkey anti-mouse (H + L) Highly Cross-Adsorbed Secondary Antibody Alexa Fluor 594 | ThermoFisher Scientific | Cat# A-21203 RRID:AB_2535789 | 1:2000 |
| antibody | Donkey anti-rabbit (H + L) Highly Cross-Adsorbed Secondary Antibody Alexa Fluor 594 | ThermoFisher Scientific | Cat# A-21207 also A21207 RRID:AB_141637 | 1:2000 |
| antibody | Donkey anti-mouse (H + L) Highly Cross-Adsorbed Secondary Antibody Alexa Fluor 488 | ThermoFisher Scientific | Cat# A-21202 RRID:AB_141607 | 1:2000 |
| antibody | Donkey anti-rabbit (H + L) Highly Cross-Adsorbed Secondary Antibody Alexa Fluor 488 | ThermoFisher Scientific | Cat# A-21206 also A21206 RRID:AB_2535792 | 1:2000 |

*Continued on next page*

*Continued*

| Reagent type (species) or resource | Designation | Source or reference | Identifiers | Additional information |
|---|---|---|---|---|
| antibody | Goat anti-Rat IgG (H + L) Cross-adsorbed Secondary Antibody Alexa Fluor 488 | ThermoFisher Scientific | Cat# A-11006 also A11006 RRID:AB_2534074 | 1:2000 |
| commercial assay or kit | Click-iT Plus EdU Alexa Fluor 488 Imaging Kit | ThermoFisher Scientific | Cat# C10637 | |
| software, algorithm | Prism 7.0 | http://www.graphpad.com/ scientific-software/prism/ | RRID:SCR_002798 | |

## Whole mount immunohistochemistry

Diaphragm, internal intercostal and levator auris longus muscles were dissected from E18.5 embryos in oxygenated L-15 media, pinned onto Sylgard-coated dissection dishes, fixed for 1.5 hr in 1% paraformaldehyde (PFA) and blocked for 1 hr in PBS with 3% BSA (Sigma IgG free) and 0.5% Triton X-100 (PBT). Primary antibodies, diluted in PBT, were force-pipetted into the tissue. The muscles were incubated overnight on an orbital shaker in a humidified chamber at 4°C. Diaphragm muscles were washed 10 times over the course of 5 hr with PBS with 0.5% Triton X-100 (PT) at room temperature and incubated in secondary antibodies, diluted in PT, overnight on an orbital shaker at 4°C in a humidified chamber. Muscles were washed 10 times over the course of 5 hr with PT at room temperature and rinsed in PBS before the muscle was whole-mounted in 50% glycerol. Muscles from at least 3 mice of each genotype were analyzed for each experiment. We restricted quantitative analysis of the diaphragm muscle to the costal region of the diaphragm muscle, excluding crural regions of the diaphragm muscle.

## Cryosection immunohistochemistry

Limb and diaphragm muscles were embedded in OCT media and frozen on a dry-ice platform. 10 μm sections, collected onto poly-L-lysine coated glass slides, were fixed in 1–4% PFA for 10 min, washed in PBS with 3% BSA (PB) 3 times for 5 min, permeabilized with PBT for 10 min, washed in PB and incubated overnight at 4°C with primary antibodies in PB (PBT for Pax7) in a humidified chamber. Sections were washed in PB 3 times for 5 min before overnight incubation at 4°C with secondary antibodies, diluted in PBS, in a humidified chamber. Sections were washed 3 times for 5 min in PB, then PBS or counterstained with DAPI, before mounting in VECTASHIELD anti-fade mounting medium.

## Cell culture immunohistochemistry

Monolayer cell cultures were plated on Collagen-coated dishes, fixed in 1–4% PFA for 10 min, permeabilized in PT for 10 min and incubated with primary antibodies in PB overnight at 4°C in a humidified chamber. Cells were washed in PB 3 times for 5 min before overnight incubation in a humidified chamber at 4°C with secondary antibodies, diluted in PBS. Cells were washed 3 times for 5 min in PB, then PBS or counterstained with Hoechst 33342 before mounting in VECTASHIELD anti-fade mounting media.

## Single fiber dissociation

Diaphragm muscles were digested with 2 mg/ml Collagenase Type Ia (Sigma) in DMEM at 37°C for 30 min on an orbital shaker. After digestion, diaphragm muscles were flushed with 5% Fetal Bovine Serum in DMEM. Diaphragm muscles were carefully dissected away from the central tendon and ribcage to ensure isolation of full-length muscle fibers, and the myofibers were dissociated with fire-polished glass Pasteur pipettes under a dissecting microscope until individual fibers were released. Single fibers adhered onto poly-L-Lysine coated coverslips were fixed in 1% PFA for 5 min and washed in PB before overnight incubation in a humidified chamber at 4°C with primary antibodies, diluted in PB. Single fibers were washed in PB and incubated in a humidified chamber overnight at 4°C in primary antibodies, diluted in PBS. Single fibers were washed and counterstained with Alexa Fluor 594 Phalloidin (ThermoFisher Scientific) before mounting in VECTASHIELD mounting medium.

## C2C12 cell cultures

C2C12 cells were propagated in growth medium (GM: 10% Fetal bovine serum in Dulbecco's Modified Eagle's Medium supplemented with 4.5 g/L glucose, L-glutamine and sodium pyruvate) at 37°C. Differentiation was induced, when cells were 80% confluent, by switching to DMEM supplemented with 2% heat-inactivated horse serum, 4.5 g/L glucose and 1 mM L-glutamine.

## Western blotting and immunoprecipitation

Cells were lysed in 30 mM triethanolamine, 1% NP-40, 50 mM NaF, 2 mM $NaV_2O_5$, 1 mM, Na-tetrathionate, 5 mM EDTA, 5 mM EGTA, 100 mM N-ethylmaleimide, 50 mM NaCl, 10 uM Pepstatin A, Roche Protease inhibitor tablet (pH 7.4) for 30 min at 4°C. Lysates were centrifuged in a microcentrifuge for 20 min at 12,000 rpm at 4°C to remove cellular debris. Protein was measured by a Bradford assay, and samples were either flash-frozen in SDS sample buffer or processed for immunoprecipitation.

Abl2 was immunoprecipitated from lysates with goat anti-Abl2 antibodies (Santa Cruz C20) and Protein-G Agarose Beads (Roche), following the manufacturer's instructions. Protein was eluted from the beads with SDS sample buffer and fractionated by SDS-PAGE. Proteins were transferred onto PVDF membranes, which were probed with antibodies to Abl2 (Rabbit polyclonal, Proteintech). We used antibodies to pan-Cadherin (Cell Signaling Technologies) and Myosin heavy chain My32 (Sigma) to ensure equal loading of proteins in whole cell lysates.

## Primary cultures and in vivo proliferation assays

Costal domains of diaphragm muscles were dissected from E18.5 embryos and separated from the central tendon in oxygenated L-15 medium under sterile conditions. Diaphragm muscles were minced and digested in 5 mg/ml Papain (Worthington Biochemicals), reconstituted in HBSS (including $CaCl_2$, $MgCl_2$ and 15 mM HEPES), following the manufacturer's instructions, for 30 min at 37°C with mild shaking. Digested cells were serially triturated with small-bore, fire-polished pipettes in DMEM/F12 with 15 mM HEPES and 20% FBS, passed through 40 μm cell strainers and plated at low densities ($5.0 \times 10^3$ cells per cm$^2$) onto Collagen-coated dishes. After 36 hr in culture, cells were treated with EdU for 1 hr and subsequently fixed in 4% PFA, permeabilized in PT, processed for primary and secondary antibody staining and processed for Click-iT EdU Plus Alexa 488 imaging, following the manufacturer's instructions (ThermoFisher Scientific). The scatter plots show the summed values from at least 150 cells in five random fields of view from individual cultures, as well as the mean ±SEM.

For in vivo EdU analysis, pregnant mice, at 13.5–14.5 days of gestation, were injected IP with 0.5 mg of EdU. Embryos were harvested 1 hr after *in utero* labeling; embryos were fixed with PFA, permeabilized in PT, stained with primary and secondary antibodies and processed for Click-iT EdU Plus Alexa 488 imaging, following the manufacturer's instructions (ThermoFisher Scientific). Confocal images were captured from 3 regions of E13.5–14.5 diaphragms, collected from at least 3 embryos of each genotype: the myotendinous region of the left hemidiaphragm, the right hemidiaphragm, and the dorsal diaphragm.

## Primary cultures and differentiation assays

Costal domains of diaphragm muscles were dissected from E18.5 embryos and separated from the central tendon in oxygenated L-15 medium under sterile conditions. Diaphragm muscles were minced and digested in 5 mg/ml Papain, reconstituted in HBSS (with $CaCl_2$ and $MgCl_2$), for 30 min at 37°C with mild shaking. Digested cells were serially triturated with small-bore, fire-polished pipettes in DMEM/F12 with 15 mM HEPES and 20% FBS, and passed through 40 μm cell strainers. Cells were re-suspended in 5% HS in DMEM and plated onto Collagen-coated dishes at $1.0 \times 10^5$ cells per cm$^2$. After differentiation, cells were fixed in 1% PFA, permeabilized with PT, and processed for primary and secondary antibody staining. Fusion indices were calculated by dividing the number of MyoD$^+$ nuclei that are contained within MHC myotubes by the total number of MyoD$^+$ nuclei.

## Imaging and data quantification

Images were acquired with a Zeiss AxioZoom stereo microscope and Zeiss LSM 700 or 800 confocal microscopes. Adjustments to detector gain and laser intensity were made to avoid saturated pixels.

Muscle length, ectopic muscle island orientation, cross-sectional area was quantified using FIJI/ImageJ software. The data in scatter plots are expressed as mean values ± SEM, unless otherwise noted in figure legends. Statistical differences between groups were evaluated using unpaired *t*-tests with Welch's correction.

### *Abl2* riboprobes

Full-length cDNA for *Abl2* was purchased from OpenBiosystems (Accession number BC065912). 5'-ACACAGGCCTCCAGTGGG-3' and 5'-GCACATCAGCTGGAGTGTGTTTC-3' primers were used to amplify a 1088 bp fragment, and 5'-AGGACCCTGAGGAAGCAGGGG-3' and 5'-TCTGTGCCAA TGAGCTGCACATC-3' primers were used to amplify a 1321 bp fragment. Amplicons were subcloned into pBluescript KS plus vectors, digested with Kpn1/Hind3 and Sac1/Hind3 restriction endonucleases, respectively. DIG RNA labeling kit (Roche) was used to prepare digoxigenin-labeled antisense and sense strand riboprobes, following the manufacturer's instructions.

## Embryonic diaphragm whole mount in situ hybridization

Diaphragm muscles were fixed overnight at 4°C in 4% PFA, dehydrated in methanol, treated with 6% $H_2O_2$ to inactivate endogenous alkaline phosphatase activity and digested for 30 min with 20 μg/ml Proteinase K. Tissues were hybridized with digoxigenin-labeled riboprobes directed against mRNA encoding 1088 bp and 1321 bp fragments of *Abl2* and processed using standard RNA in situ hybridization protocols (*Kim and Burden, 2008*).

## Electron microscopy

Adult diaphragm muscles were treated with 1 μg/ml Tetrodotoxin and fixed in 1% glutaraldehyde. Fixed muscles where treated with 1% osmium for 1 hr, stained en bloc with saturated aqueous uranyl acetate for 1 hr, dehydrated, and embedded in Epon (*Friese et al., 2007*).

## Behavior experiments

Running endurance was performed utilizing wireless running wheels and analysis software obtained from Med Associates Inc with age-matched female littermates at 18 weeks old maintained in a C57BL/6 genetic background. Mice were maintained in 12 hr light dark cycles and individual mice were placed in their home cage with one running wheel per cage, three hours prior to lights out on experimental days. Basal breathing analysis was performed using the Buxco Finepointe whole body plethysmography system. Grip force measurements were performed using a digital force gauge.

## Generation of *Abl2* flox mice

Knockout first alleles of *Abl2* were generated as part of the International Knockout Mouse Consortium by IMPC/KOMP/EUCOMM (*Skarnes et al., 2011*). Targeted ES cells were obtained and injected into B6-albino blastocysts. Chimeric mice were backcrossed to C57BL/6J mice and selected for germline transmission yielding 'knockout-first' *Abl2-tm1a* mice that possess an interrupting *LacZ* element, flanked by FRT sites, between exons 5 and 6 that serves as both a reporter and null allele. 'Knockout-first' *Abl2-tm1a* mice were bred to mice that express Flp recombinase under the control of Pgk1 promoter (FLPo-10 Jax stock number 011065) to remove FRT flanked *LacZ* cassettes, yielding an allele that possesses *LoxP* sites flanking exon 6, e.g. *Abl2* 'flox' *Abl2-tm1c* mice.

### *LacZ* genetic labeling and whole mount X-Gal staining

We crossed $Pax3^{cre/+}$; $Abl2^{+/-}$ mice with $Abl2^{+/-}$; $ROSA26^{LacZ/+}$ mice to generate (1) $Pax3^{cre/+}$; $ROSA26^{LacZ/+}$, (2) $Abl2^{-/-}$; $Pax3^{cre/+}$; $ROSA26^{LacZ/+}$ and (3) $Abl2^{+/-}$; $Pax3^{cre/+}$; $ROSA26^{LacZ/+}$ embryos. E12.5-E14.5 embryos were stained for β-galactosidase activity. After overnight fixation at 4°C with 4% PFA (in 0.1M PIPES, 0.2 mM $MgCl_2$, 0.125M EGTA pH 6.9), embryos were washed twice for 30 min (in PBS with 2 mM $MgCl_2$). Following removal of thoracic organs, the embryos were permeabilized overnight at 4°C with 0.1% NP40 and 0.01% deoxycholate (in PBS with 2 mM $MgCl_2$,) and stained overnight at 37°C to detect β-galactosidase activity (1 mg/ml X-gal in the permeabilization buffer with 17.5 mM $K_3Fe(CN)_6$ and 17.5 mM $K_4Fe(CN)_6$). Embryos were rinsed (in PBS with 2 mM MgCl2), post-fixed with 4% PFA, and cleared with 80% glycerol before imaging.

## Antibodies used for IHC/IFC

We used the following primary antibodies: Sigma mouse anti-My-32 (1:1000), Sigma mouse anti-NOQ7.5.4D myosin heavy chain slow (1:1000), Sigma Rabbit anti-Laminin (1:1000), Santa Cruz mouse anti-Pax7 DSHB bioreactor supernatant (1:500), Santa Cruz Rabbit anti-MyoD C-20 (1:1000), Thermofisher Mouse anti-MyoD 5.8A.

We used the following secondary antibodies: ThermoFisher Scientific Donkey anti-mouse (H + L) Highly Cross-Adsorbed Secondary Antibody Alexa Fluor 594 or Alexa Fluor 488, Donkey anti-rabbit (H + L) Highly Cross-Adsorbed Secondary Antibody Alexa Fluor 594 or Alexa Fluor 488, Goat anti-Rat IgG (H + L) Cross-adsorbed Secondary Antibody Alexa Fluor 488, Life Technologies.

## Acknowledgements

We are grateful to Dr. Sang Yong Kim, director of the Rodent Genetic Engineering Core at NYUMC, for his assistance in deriving the *Abl2flox* mice described here. This work was supported with funds from the NIH (NS36193 and NS075124). We thank Maartje Huibers and Julien Oury for their comments on the manuscript.

## Additional information

### Funding

| Funder | Grant reference number | Author |
| --- | --- | --- |
| National Institute of Neurological Disorders and Stroke | R37NS36193 | Steven J Burden |
| National Institute of Neurological Disorders and Stroke | RO1NS075124 | Steven J Burden |

The funders had no role in study design, data collection and interpretation, or the decision to submit the work for publication.

### Author contributions

Jennifer K Lee, Conceptualization, Data curation, Formal analysis, Investigation, Methodology, Writing—original draft, Writing—review and editing; Peter T Hallock, Conceptualization, Investigation, Methodology, Writing—review and editing; Steven J Burden, Conceptualization, Supervision, Funding acquisition, Writing—original draft, Project administration, Writing—review and editing

### Author ORCIDs

Steven J Burden [iD] https://orcid.org/0000-0002-3550-6891

### Ethics

Animal experimentation: Animal experimentation: All procedures were approved the New York University School of Medicine Institutional Animal Care and Use Committee (Protocol 160425).

### Decision letter and Author response

Decision letter https://doi.org/10.7554/eLife.29905.018
Author response https://doi.org/10.7554/eLife.29905.019

## Additional files

### Supplementary files

• Transparent reporting form
DOI: https://doi.org/10.7554/eLife.29905.016

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
