## [Decision Letter]

Thank you for submitting your article "Abelson tyrosine-protein kinase 2 Regulates Myoblast Proliferation and Controls Muscle Fiber Length" for consideration by *eLife*. Your article has been reviewed by three peer reviewers, and the evaluation has been overseen by a Reviewing Editor and Didier Stainier as the Senior Editor. The following individual involved in review of your submission has agreed to reveal her identity: Gabrielle Kardon (Reviewer #2).

The reviewers have discussed the reviews with one another and the Reviewing Editor has drafted this decision to help you prepare a revised submission.

Summary:

An important structure/function characteristic of muscles is their size. Individual myofibers are characterized by their cross-sectional area and their length. While the cross-sectional area of myofibers can change in the adult (e.g. during hypertrophy), their length is generally thought to be fixed. Surprisingly, how myofiber length is regulated is unknown. This paper identifies that the tyrosine protein kinase 2 Abl2 regulates myofiber length in three muscles – diaphragm, intercostal muscles, and levator auris longis. They find that Abl2 acts cell autonomously in myoblasts to negatively regulate levels of proliferation and this, in turn, regulates myofiber length in these muscles. Overall, this is an interesting paper that offers the first insights into how muscle fiber length is controlled on a molecular and cellular level.

Essential revisions:

1) The authors need to provide more analysis at the cellular level to explain why fiber length and not diameter of diaphragm myofibers affected.

Is there an increase in number of muscle progenitors or myoblasts in Abl2-/-mutants, in vitro and in vivo? It is assumed that the increase of muscle fiber length is due to an increase of MyoD+ cell proliferation and then fusion, which is recognized to occur at muscle extremities. Does the increase in the number of MyoD+ cells occur only at the extremities of muscle fibers? Is Abl2 expression regionalized in a subpopulation of myoblasts within muscles. The fusion index should be recalculated as the number of myonuclei over the total number of nuclei.

2) Do cultures of Abl2-/- give any insights? Are myofibers from cultured Abl2-/- myoblasts longer than controls? in vivo or in vitro, is the number of myonuclei/unit of myofiber length increased or decreased in Abl2-/- myofibers? Is there a difference in the myonuclei distribution along the myofiber? This data should be straightforward to obtain with the myoblast cultures and myofiber preps already in hand. It is important to provide more mechanistic insight into what is different in the longer Abl2-/- myofibers.

3) Why are only a limited number of muscles affected?

Why would the increase of myoblast proliferation not occur in limb muscles depleted in Abl2? Is Abl2 only expressed in these muscle regions? Single fiber or myoblast cultures from limb muscles would be more convincing than global length analysis of TA muscles (Figure 2—figure supplement 1).

4) Abl2+/- versus Abl2-/- phenotypes: Abl2+/- mice displayed ectopic muscle islands in the tendon region of the diaphragm. Both phenotypes (increased muscle fiber length and ectopic muscle islands) are striking and potentially interesting, albeit difficult to reconcile with each other. There is no obvious attempt in the MS to explain the link between the phenotypes. The authors provide convincing data the islands are not formed from tendon precursors, thus increasing the likelihood it is a myogenic-driven phenotype. It is important to provide more insight at the level of cellular mechanism into the difference between losing one or both copies of Abl2.

For example, it is not mentioned if there is any change in muscle fiber length or myoblast proliferation in Abl2+/- mice. Are there any ectopic muscles in tendon regions like those in the diaphragm of Abl2+/- mice? The data showing that ectopic muscle islands are also observed in conditional Abl2+/- mice (Pax3 and MyoD + Cells) are important and should be shown and not only mentioned as "data not shown" in the Discussion section.

5) What happens to Scleraxis expression in axial tendons? Better characterization of the timing of the formation of muscle fibers and somite boundaries would provide critical information for understanding the underlying mechanisms leading to the long muscle fibers. This phenotype of long fibers crossing the borders is reminiscent of that of Notch mutants in zebrafish (Henry et al., 2005). Moreover, in zebrafish, muscles are important for myoseptum development (Tom Schilling's work). How does Scleraxis expression relate to Abl expression in different muscles?

6) Anatomical variation: The diaphragm muscle is composed of two muscle domains, the costal diaphragm which is a thin sheet of radially arrayed myofibers and the crural diaphragm which is thicker and positioned more posteriorly/dorsally. The authors do not consider these domains or how it may influence the acquisition, analysis, and interpretation of the dataset. As an example, there is a range of myofiber lengths in control costal and crural muscles by visual inspection. In the abl2 mutants, the range seems to disappear, giving a uniform "long" (Figure 1); but this isn't reflected in the number of myonuclei added (more of a range; Figure 3). Does this have something to do with these domains and data acquisition? We request that the authors attempt to stratify their data based on the domains or explicitly state the domain they are analyzing in each experiment.

---

## [Author Response]

Essential revisions:1) The authors need to provide more analysis at the cellular level to explain why fiber length and not diameter of diaphragm myofibers affected.

*Is there an increase in number of muscle progenitors or myoblasts in Abl2-/-mutants,* in vitro *and* in vivo*? It is assumed that the increase of muscle fiber length is due to an increase of MyoD+ cell proliferation and then fusion, which is recognized to occur at muscle extremities. Does the increase in the number of MyoD+ cells occur only at the extremities of muscle fibers? Is Abl2 expression regionalized in a subpopulation of myoblasts within muscles. The fusion index should be recalculated as the number of myonuclei over the total number of nuclei.*

We found no steady state increase in the number of MyoD+ cells isolated from E18 *abl2* mutant mice. We have added these data to Figure 6 and discussed these data in the text. We believe that these data are consistent with the idea that the increased proliferation rate of *abl2* mutant MyoD+ cells, documented in vivoand in vitro(Figure 6), is accompanied and balanced by rapid fusion of these myoblasts to form syncytial myotubes.

Our data demonstrate a role for Abl2 in myoblast proliferation and do not support a direct role for Abl2 in myoblast fusion. As such, we do not wish to invoke a novel mechanism for fusion of abl2 mutant myoblasts during development and therefore cited one of several studies (Zhang and McLennan, 1995), demonstrating that myoblast fusion occurs at the growing ends of developing muscle. To determine whether abl2 mutant myoblasts violate this rule and fuse randomly along a developing myotube would be a study in itself and require substantial time and effort, which we believe is not justified.

We have no reason to suspect that Abl2 expression is restricted to a subpopulation of myoblasts. Moreover, we tested available antibodies to Abl2 by staining wild-type and abl2 mutant cells in culture and found that available antibodies to Abl2 stained abl2 mutant cells and therefore are neither specific for Abl2 nor reliable for identifying Abl2‐expressing cells in tissue.

We studied primary muscle cultures and not muscle cell lines to measure the fusion index. The fusion index is often calculated by measuring the number of nuclei in myotubes divided by the total number of nuclei in the culture dish. This procedure is meaningful when studying a cell line that contains only myoblasts. However, when studying primary cultures, in which myoblasts and non-muscle cells are nearly equally abundant, including non-muscle cells in this calculation, obfuscates the measure of fusion. Instead, we used a procedure that counts the number of nuclei in myotubes divided by the number of nuclei in myotubes plus myoblasts, marked by MyoD expression, excluding non-muscle cells, which we believe provides a more insightful calculation of myoblast fusion in primary cultures.

In cell culture, myoblasts fuse promiscuously with myotubes all along their length, and myotubes fuse haphazardly with one another, leading to branched myotubes and myotubes with unusual shapes that do not resemble myotubes found in vivo. We quantified the number of nuclei per myotube (Figure 6), which we believe is a more meaningful assessment of myoblast fusion in cell culture rather than measuring the shape/length of myotubes in vitro.

We agree that it is important and interesting to know whether the distribution of myonuclei is altered in lengthened, abl2 mutant myofibers. We presented data showing that the myonuclei distribution in lengthened myofibers was not altered: the number of myonuclei/unit of myofiber length was unchanged

(see graph in Figure 3).

*2) Do cultures of Abl2-/- give any insights? Are myofibers from cultured Abl2-/- myoblasts longer than controls?* in vivo *or* in vitro*, is the number of myonuclei/unit of myofiber length increased or decreased in Abl2-/- myofibers? Is there a difference in the myonuclei distribution along the myofiber? This data should be straightforward to obtain with the myoblast cultures and myofiber preps already in hand. It is important to provide more mechanistic insight into what is different in the longer Abl2-/- myofibers.*

The reviewers are intrigued that diaphragm, intercostal and levator auris muscles are lengthened, due to increased myoblast proliferation in the absence of Abl2, whereas limb muscle length is unaffected. To determine whether limb muscles had accrued more nuclei but had hypertrophied rather than lengthened, we measured muscle fiber diameter. Figure 2—figure supplement 1 shows that the cross-sectional area of limb muscle fibers, like diaphragm muscle fibers (Figure 2) was not changed, inconsistent with the idea that limb muscle fibers had hypertrophied and increased in diameter instead of lengthened. We agree that understanding how limb muscle development proceeds in the absence of Abl2 is interesting, but we don’t believe that this knowledge would provide information that is essential to justify the conclusions of this study and would instead extend the study in a new direction.

3) Why are only a limited number of muscles affected?Why would the increase of myoblast proliferation not occur in limb muscles depleted in Abl2? Is Abl2 only expressed in these muscle regions? Single fiber or myoblast cultures from limb muscles would be more convincing than global length analysis of TA muscles (Figure 2—figure supplement 1).

As requested by the reviewers, we have now included data showing that abl2 ^+/-^ myoblasts proliferate at a rate that is intermediate between wild‐type myoblasts and abl2 ^‐/-^ myoblasts (Figure 6), and we have discussed these data in the discussion. These findings are consistent with the idea that increased proliferation of abl2 ^+/-^ myoblasts contributes to the formation of ectopic muscle islands.

We agree that it is important to include data showing ectopic islands in muscle‐conditional abl2 heterozygous mice. We have now included these data in Figure 8.

The large and flat central tendon of the diaphragm presents a unique opportunity to visualize the ectopic muscle islands. We are not aware of other muscles that present such a favorable opportunity.

We provide data showing that ectopic muscle islands are not formed from tendon precursors (Figure 8). In addition, we analyzed conditional mutant mice (Pax3cre; abl2f/+), which demonstrate that the formation of ectopic muscle islands depends upon the loss of Abl2 from the muscle lineage (Figure 8).

Although it would be interesting to know more about the mechanisms responsible for the formation of the ectopic muscle islands, the conclusions that we draw from the presence and arrangement of the ectopic islands in the diaphragm muscle stand on their own and do not require knowledge of how the islands arose in the first place. Independent of understanding precisely how the muscle islands form, the common length and orientation of the myofibers in the ectopic islands, which are surrounded by tendon cells, excludes certain mechanisms for establishing muscle fiber length and orientation and clearly reveals that tendon differentiation can be controlled by muscle.

4) Abl2+/- versus Abl2-/- phenotypes: Abl2+/- mice displayed ectopic muscle islands in the tendon region of the diaphragm. Both phenotypes (increased muscle fiber length and ectopic muscle islands) are striking and potentially interesting, albeit difficult to reconcile with each other. There is no obvious attempt in the MS to explain the link between the phenotypes. The authors provide convincing data the islands are not formed from tendon precursors, thus increasing the likelihood it is a myogenic-driven phenotype. It is important to provide more insight at the level of cellular mechanism into the difference between losing one or both copies of Abl2.For example, it is not mentioned if there is any change in muscle fiber length or myoblast proliferation in Abl2+/- mice. Are there any ectopic muscles in tendon regions like those in the diaphragm of Abl2+/- mice? The data showing that ectopic muscle islands are also observed in conditional Abl2+/- mice (Pax3 and MyoD + Cells) are important and should be shown and not only mentioned as "data not shown" in the Discussion section.

We have now presented data showing the stage during development when an increase in muscle fiber length in abl2 mutant mice is first detectable (Figure 1—figure supplement 1).

5) What happens to Scleraxis expression in axial tendons? Better characterization of the timing of the formation of muscle fibers and somite boundaries would provide critical information for understanding the underlying mechanisms leading to the long muscle fibers. This phenotype of long fibers crossing the borders is reminiscent of that of Notch mutants in zebrafish (Henry et al., 2005). Moreover, in zebrafish, muscles are important for myoseptum development (Tom Schilling's work). How does Scleraxis expression relate to Abl expression in different muscles?

We restricted our analysis of the diaphragm muscle to the costal diaphragm muscle, and we have now stated this explicitly in the text.

6) Anatomical variation: The diaphragm muscle is composed of two muscle domains, the costal diaphragm which is a thin sheet of radially arrayed myofibers and the crural diaphragm which is thicker and positioned more posteriorly/dorsally. The authors do not consider these domains or how it may influence the acquisition, analysis, and interpretation of the dataset. As an example, there is a range of myofiber lengths in control costal and crural muscles by visual inspection. In the abl2 mutants, the range seems to disappear, giving a uniform "long" (Figure 1); but this isn't reflected in the number of myonuclei added (more of a range; Figure 3). Does this have something to do with these domains and data acquisition? We request that the authors attempt to stratify their data based on the domains or explicitly state the domain they are analyzing in each experiment.

We have made adjustments to the Figure Legends and supplied additional detail about experimental methods there as well as in the Materials and methods section.

We have also added information to the Materials and methods section, as requested by the reviewers. We believe that the title, “Abelson tyrosine‐protein kinase 2 Regulates Myoblast Proliferation and Controls Muscle Fiber Length”, indicates that Abl2 regulates both myoblast proliferation and muscle fiber length and therefore accurately reflects the data and findings.